# Characterization of a new *Leishmania major* strain for use in a controlled human infection model

Helen Ashwin[1,7], Jovana Sadlova [2,7], Barbora Vojtkova[2], Tomas Becvar[2], Patrick Lypaczewski [3], Eli Schwartz[4], Elizabeth Greensted[1], Katrien Van Bocxlaer[1], Marion Pasin[5], Kai S. Lipinski[5], Vivak Parkash [1], Greg Matlashewski [3], Alison M. Layton[1], Charles J. Lacey [1], Charles L. Jaffe[6✉], Petr Volf [2✉] & Paul M. Kaye [1✉]

Leishmaniasis is widely regarded as a vaccine-preventable disease, but the costs required to reach pivotal Phase 3 studies and uncertainty about which candidate vaccines should be progressed into human studies significantly limits progress in vaccine development for this neglected tropical disease. Controlled human infection models (CHIMs) provide a pathway for accelerating vaccine development and to more fully understand disease pathogenesis and correlates of protection. Here, we describe the isolation, characterization and GMP manufacture of a new clinical strain of *Leishmania major*. Two fresh strains of *L. major* from Israel were initially compared by genome sequencing, in vivo infectivity and drug sensitivity in mice, and development and transmission competence in sand flies, allowing one to be selected for GMP production. This study addresses a major roadblock in the development of vaccines for leishmaniasis, providing a key resource for CHIM studies of sand fly transmitted cutaneous leishmaniasis.

[1] York Biomedical Research Institute, Hull York Medical School, University of York, York, UK. [2] Department of Parasitology, Faculty of Science, Charles University, Viničná 7, Prague, Czech Republic. [3] Department of Microbiology and Immunology, McGill University, Montreal, Quebec, Canada. [4] The Center for Geographic Medicine and Tropical Diseases, Chaim Sheba Medical Center, Sackler School of Medicine, Tel Aviv University, Tel Aviv, Israel. [5] Vibalogics GmbH, Cuxhaven, Germany. [6] The Hebrew University-Hadassah Medical School, Jerusalem, Israel. [7] These authors contributed equally: Helen Ashwin, Jovana Sadlova. ✉email: cjaffe@mail.huji.ac.il; volf@cesnet.cz; paul.kaye@york.ac.uk

The leishmaniases represent a group of diseases caused by infection with various species of the parasitic protozoan *Leishmania*. One billion people are at risk of infection across 98 countries worldwide, with over 1.5 million new cases and 20,000–40,000 deaths reported each year[1,2]. The leishmaniases are vector-borne diseases, each parasite species having co-evolved for transmission by one or more species of phlebotomine sand fly[3,4]. Disease manifestation is intimately linked to the species of infecting parasite[5,6] and may be evident as self-healing lesions restricted to the site of skin transmission (cutaneous leishmaniasis; CL), lesions which spread from an initial skin lesion to involve the mucosae (mucosal leishmaniasis; ML) or which spread uncontrolled across the body (disseminated or diffuse cutaneous leishmaniasis; DCL), or as a potentially fatal systemic disease involving major organs such as the spleen, liver and bone marrow (kala azar or visceral leishmaniasis; VL)[5]. In addition, patients recovering from VL following chemotherapy often develop a chronic skin condition (post kala-azar dermal leishmaniasis; PKDL) that can sustain community transmission of VL[7,8]. Collectively, the tegumentary forms of leishmaniasis account for approximately two-thirds of the global disease burden, while VL accounts for most reported deaths[1,6]. In addition to the impact of primary disease, recent studies have also emphasized the importance of considering the long term sequelae of leishmaniasis, notably those associated with stigmatization, when evaluating global burden of these diseases[9–11].

The leishmaniases are widely regarded as vaccine-preventable diseases based on disease natural history, epidemiological data and studies in experimental models of leishmaniasis (reviewed in refs. [12–15]). Four vaccines for canine visceral leishmaniasis have reached the market, though with remaining questions about the extent of clinical versus parasitological protection that they provide[16]. To date, no human vaccines have achieved licensure[17]. Often cited barriers to vaccine development include limited investment, with only $3.7 M of new R&D funding globally in 2018[18], an excess of candidate antigens and delivery systems[19], the questionable predictive capacity of pre-clinical animal models[20,21], lack of good correlates of protection and a defined target product profile[22], the costs and challenge of large-scale efficacy studies in disease endemic countries coupled with the high prevalence of asymptomatic infections[23,24] and a fragmented pipeline for translational research[25]. As has been found with other diseases[26–33], the incorporation of a controlled human infection model (CHIM) into the vaccine R&D pipeline can overcome many of these issues.

Artificial human infection ("leishmanization") with *Leishmania* had been practiced for centuries by people living in the Middle East and former Soviet states, where CL is highly endemic. Building on the knowledge that cure from CL engendered protection against reinfection, scrapings from active lesions were used to cause disease at a site of choice (e.g. the buttock), so avoiding the stigmatization associated with CL scars (reviewed in ref. [34]). More defined experimental studies were conducted sporadically through the 20th century, culminating in a WHO-sponsored evaluation of the potential for human challenge as a tool to evaluate vaccines for leishmaniasis, conducted in Iran in 2005[35]. This study employed a *L. major* strain that had been produced at GMP and used for previous leishmanization studies. Results from this study demonstrated a take rate of 86% in previously non-exposed volunteers. Lesions were <3 cm diameter and ulcerated in 74% of cases. All lesions self-healed without treatment between 75 and 285 days after inoculation. In a limited re-challenge study using the same strain, 0/11 volunteers receiving leishmanization developed a lesion, compared to 5/5 in non-leishmanized controls[35]. Poor viability of the challenge agent and limited funding opportunities curtailed this programme before it

could be developed further and to date, no defined vaccines have been tested using this approach.

With the advent of new candidate vaccines in or approaching the clinic, there is renewed imperative to develop a CHIM for leishmaniasis. An adenoviral-vectored vaccine (ChAd63-KH) was found to be safe and immunogenic in healthy volunteers[36] and in PKDL patients (Younnis et al. submitted) and is currently in Phase IIb as a therapeutic in Sudanese PKDL patients. A live genetically attenuated *L. donovani* centrin$^{-/-}$ parasite has shown efficacy in pre-clinical models[37–39] and a *L. major* centrin$^{-/-}$ [40] is soon to enter GMP production. An adjuvanted recombinant polyprotein vaccine (LEISH-F3/GLA-SE) has been progressed to Phase I[41] and a newer derivative (LEISH-F3+/GLA-SE) evaluated in pre-clinical models[21]. RNA-based vaccines are also in development[42]. In addition, new knowledge regarding the integral nature of sand fly transmission to *Leishmania* infectivity has emerged in recent years[43–45], underpinning the observation that vaccines inducing protection in mice when infected via needle inoculation fail to protect against sand fly-transmitted infection[46]; hence the need to incorporate vector transmission as part of a CHIM.

The pathway for development of a CHIM for sand fly-transmitted leishmaniasis requires three enabling activities: the identification of an appropriate challenge agent, optimization of sand fly transmission studies in humans, and patient and public involvement (PPI). Here, we describe completion of the first of these steps, namely the isolation, characterization and GMP production of a new *L. major* challenge agent.

## Results

**New clinical strains of *Leishmania major*.** *Leishmania major* is endemic in Israel and cases are often associated with travelers visiting areas of high transmission[47,48]. Two individuals from non-endemic areas of central Israel that had self-referred to Sheba Hospital in early 2019 after developing lesions subsequent to visiting the endemic region of Negev (Fig. 1a) served as parasite donors. Donor MRC-01 was a 41-year-old female who developed a lesion near her lip ~3 months after spending one night outdoors. She had self-administered topical antibiotics without effect and presented at clinic ~4 months later with a single erythematous 1.5 cm diameter lesion. Diagnosis for *L. major* was confirmed by PCR and she was treated with intra-lesional sodium stibogluconate (SSG) on two occasions ~4 weeks apart. Her lesion fully resolved with minimal scarring by 3 months post treatment onset (Fig. 1b). Donor MRC-02 was a 22-year-old male who developed two papules on his shin ~2 months after hiking in the Negev. He attended clinic 3 months later with two ~1.5 cm diameter ulcerated lesions on the shin and a very small non-ulcerated lesion on his neck. He was diagnosed positive for *L. major* by PCR but refused treatment. His lesions fully resolved ~3–4 months later with scarring (Fig. 1c). Both donors were negative for HIV, HTLV-1, HBV and HCV, and at 18-month follow-up, neither reported any reactivation of their lesion(s) or other unexpected clinical events related to their leishmaniasis.

**Whole-genome sequencing of parasite strains from patients MRC-01 and MRC-02.** Parasites were isolated from slit skin preparations and diagnosis of *L. major* infection confirmed by PCR and RFLP analysis. The strains were designated as *L. major* MHOM/IL/2019/MRC-01 and *L. major* MHOM/IL/2019/MRC-02 (herein referred to as *L. major* MRC-01 and *L. major* MRC-02, respectively). Parasites were minimally cultured in GMP grade media to retain infectivity and multiple vials frozen at P1 as a seed bank and screened negative for mycoplasma. The seed stock was redistributed under dry ice using a commercial shipping agent.

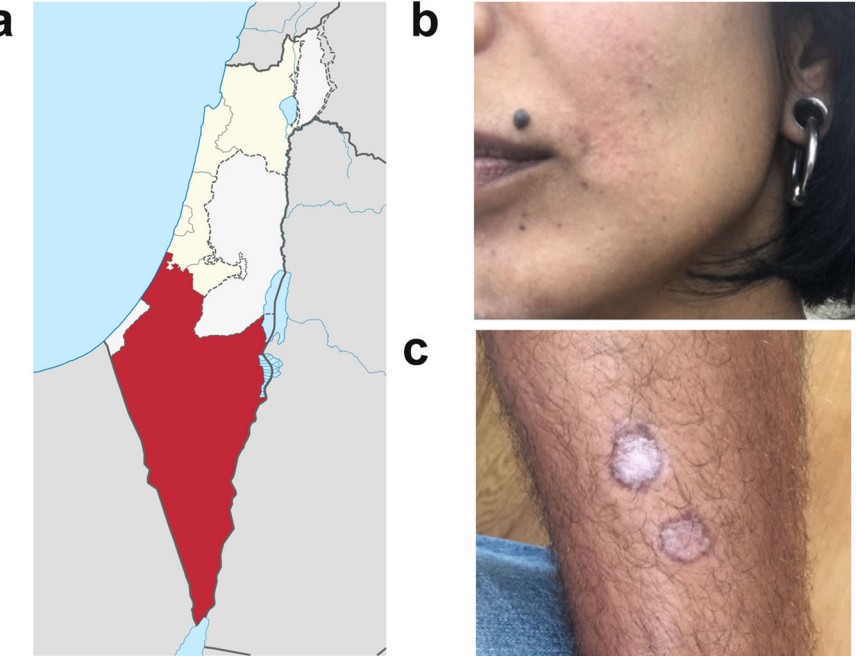

**Fig. 1 Clinical characteristic of patient lesions. a** Negev region of Israel (red). Source: Israel location map.svg NordNordWestderivative work: ויקיג'אנקי, CC BY-SA 3.0 https://creativecommons.org/licenses/by-sa/3.0, via Wikimedia Commons. **b** Donor MRC-01 lesion photographed ~3 months after treatment onset, showing full resolution of the lesion near the corner of the mouth, with minimal scarring. **c** Donor MRC-02 lesions photographed ~9 months after travel to the endemic region, showing healing without treatment.

To confirm genetic identity and establish baseline sequence data, whole-genome sequencing was performed using Illumina NextSeq deep sequencing. Sequence data for *L. major* MRC-01 and *L. major* MRC-02 has been deposited at GenBank. A phylogeny tree was developed using all available whole-genome *L. major* sequences from around the world with *L. major* Friedlin strain from Israel used as reference (Fig. 2a). This analysis shows a geographical clustering of genome sequences and confirms that *L. major* MRC-01 and *L. major* MRC-02 are closely related to other *L. major* strains derived from Israel while being distinct from each other. Next, sequence alignment was performed to identify the location of single nucleotide polymorphisms relative to the reference Friedlin strain, using DNA from early culture passage *L. major* MRC-01 and *L. major* MRC-02 (grey lines, Fig. 2b). Both strains had a number of homozygous SNPs compared to the reference Friedlin strain in all 36 chromosomes (Fig. 2b, inner two rings). Similarly, the SNP fingerprint of *L. major* MRC-01 compared to *L. major* MRC-02 also reveal these are genetically distinct strains consistent with the phylogeny tree.

We also compared whole-genome sequences from *L. major* MRC-02 before and after a single passage in BALB/c mice (see below). Compared to early culture parasites (pre-infection), the SNP fingerprint of the BALB/c passaged parasites (post-infection) was nearly identical (Fig. 2b, outer two rings). Only three additional polymorphisms were identified in parasites recovered following in vivo passage, and all were heterozygous SNPs in homopolymer stretches in non-coding regions (Supplementary Fig. 1a, b). In addition, we found no significant copy number variation (CNV) differences at the gene or chromosome level between the genomes from *L. major* MRC-02 (pre, culture only) and *L. major* MRC-02 (post, in vivo infection) parasites (Supplementary Fig. 2). These observations confirm that *L. major* MRC-01 and *L. major* MRC-02 are closely related to strains previously isolated in Israel, are genetically distinct from each other and that there was no selection for genomic mutations or CNVs following *L. major* MRC-02 infection in BALB/c mice.

*Leishmania* RNA viruses (LRV) have been demonstrated in various *Leishmania* species: LRV1 in *L.* (*Viannia*) *braziliensis* and *L.* (*V.*) *guyanensis*, and LRV2 in *L. aethiopica*, *L. infantum*, *L. major* and *L. tropica*. RNA was isolated from early passage *L. major* MRC-01 and *L. major* MRC-02, and tested for LRV2 by RT-PCR[49]. *L. aethiopica* LRC-L494, previously shown to contain LRV2, was used as a positive control. Both *L. major* strains were negative for LRV2 (Supplementary Fig. 3).

**In vitro and in vivo characterization and drug sensitivity of *L. major* MRC-01 and *L. major* MRC-02.** Prior to in vivo infectivity studies, each strain was evaluated for growth under standard in vitro conditions. Both strains showed similar in vitro growth curves, with characteristic progression through logarithmic and stationary phases of growth (Fig. 3a). Metacyclic promastigotes (Fig. 3b) were isolated by negative selection using PNA[50] and used for infectivity studies in mice. To confirm in vivo infectivity and assess sensitivity to paromomycin (PM), a standard drug used for the treatment of CL[51], we used susceptible BALB/c mice infected subcutaneously in the rump with ~$10^6$ purified metacyclic promastigotes (Figs. 3 and 4). Mice were randomized to receive PM (50 mg/kg i.p. daily for 10 days), with treatment starting when individual lesion size was 3–4 mm in diameter. Both strains induced lesions in BALB/c mice (Fig. 3c, f). In the absence of treatment, all infected mice progressed to the pre-determined endpoint (8 mm average diameter, <10 mm in any direction) or showed progressing disease at the experimental endpoint of 70 days post infection. However, *L. major* MRC-02 lesions developed more rapidly and in a more consistent manner. For example, median time to develop a lesion >2 mm was 37.5 vs. 21.0 days, for *L. major* MRC-01 and *L. major* MRC-02, respectively (ratio 1.786, 95% CI of ratio 0.96 to 3.32; $p < 0.0001$, Fig. 3d, g and Supplementary Fig. 4). All mice responded well to PM treatment, with a reduction in lesion size that often reached the limits of detection within the 10-day treatment window (Figs. 3e,

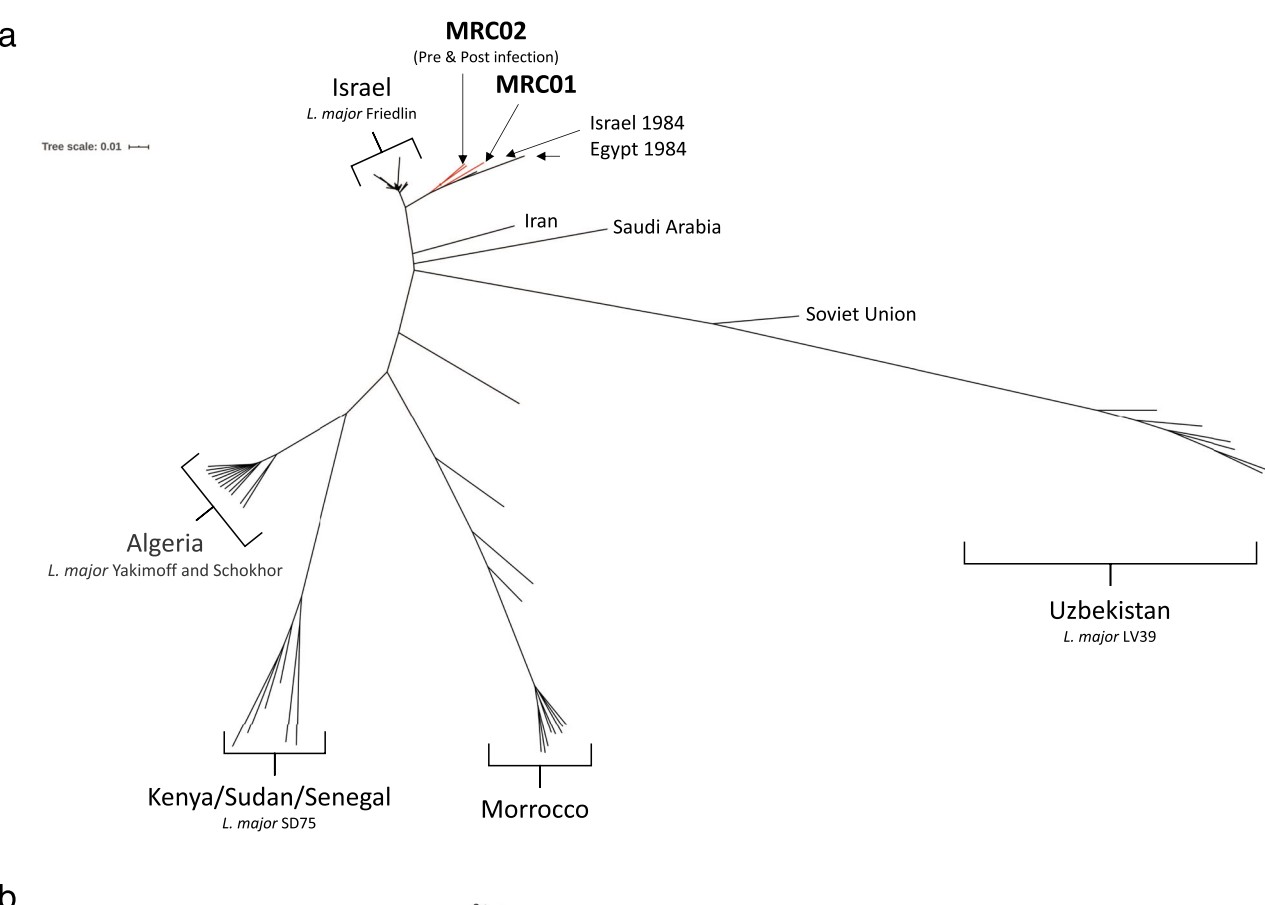

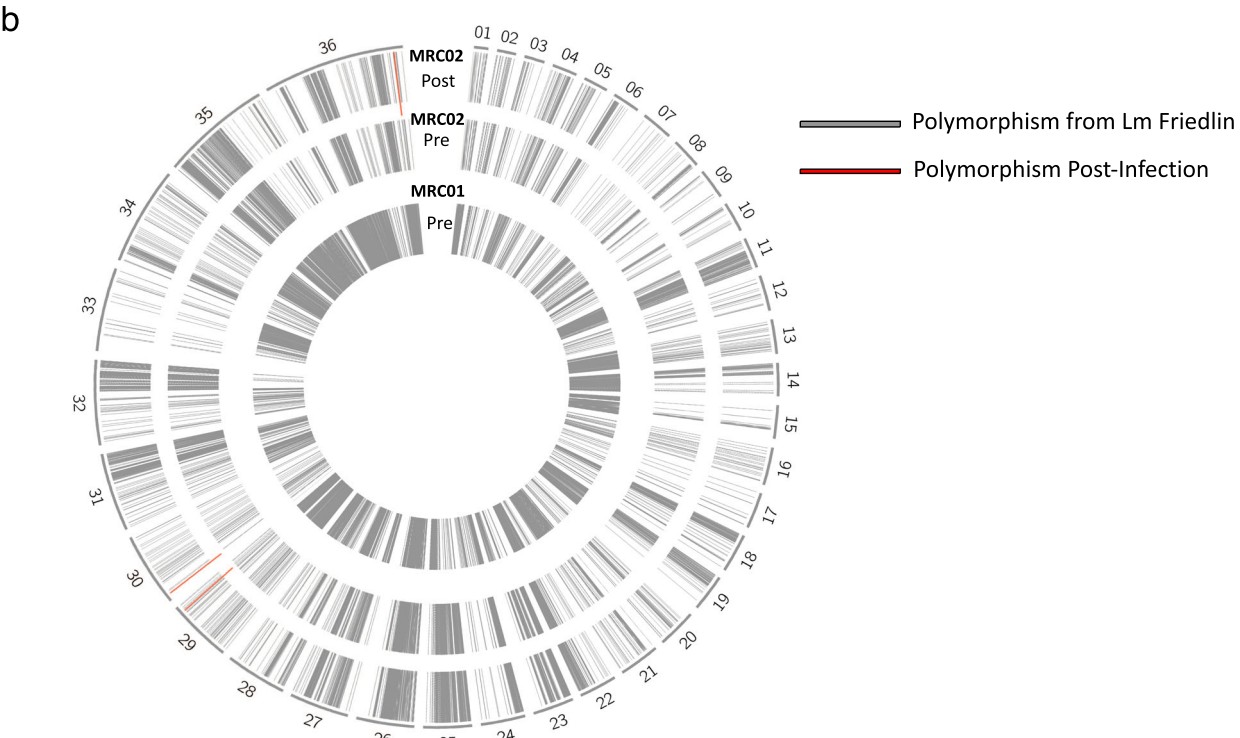

**Fig. 2 Characterisation of *L. major* MRC-01 and *L. major* MRC-02 by whole-genome sequencing. a** Phylogeny tree developed using all available whole-genome sequences for *L. major* from different parts of the world and the location of the *L. major* MRC-01 and *L. major* MRC-02 strains. The phylogeny tree was constructed using the *L. major* Friedlin strain as the reference strain. **b** Alignment map for *L. major* chromosomes 1–36. Grey bars represent location of homozygous single nucleotide polymorphism (SNPs) and indels where there are differences between *L. major* MRC-01, *L. major* MRC-02 and *L. major* MRC-02 after passage infection in BALB/c mice) and the reference strain *L. major* Friedlin. Red bars indicate chromosomal location of SNPs/indels differences between *L. major* MRC-02 (pre-infection) and the *L. major* MRC-02 (post-infection).

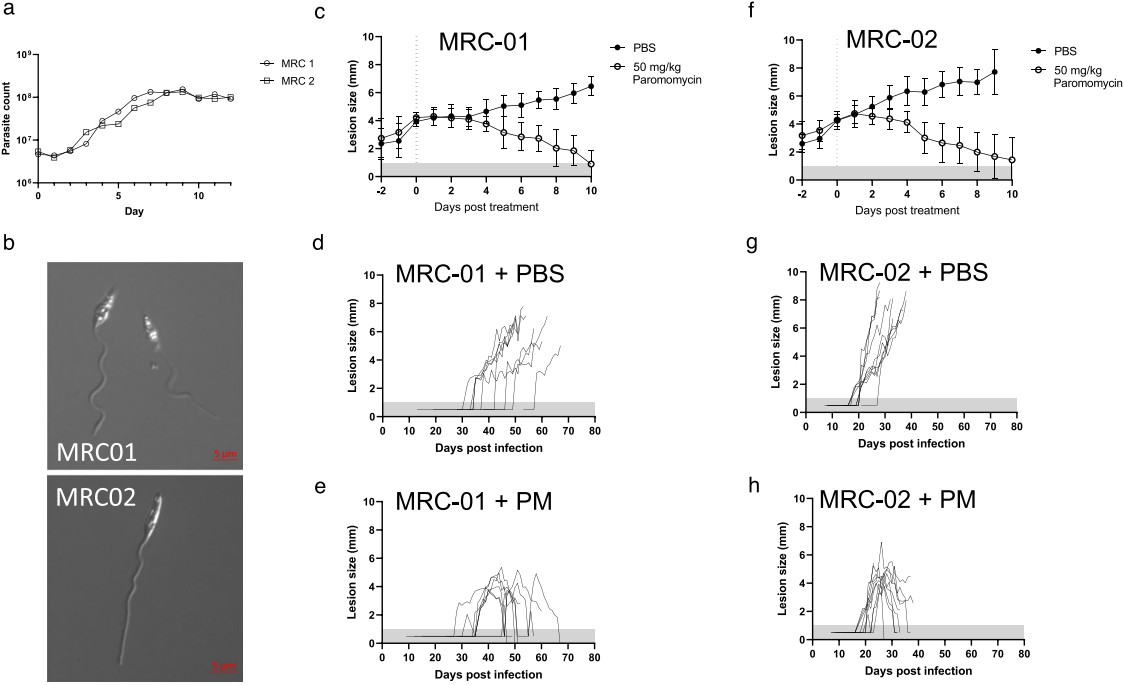

**Fig. 3 Growth characteristics and drug sensitivity of *L. major* MRC-01 and *L. major* MRC-02. a** Growth curves for *L. major* MRC-01 and *L. major* MRC-02 in vitro. Representative of one experiment from two performed. **b** Photomicrographs of purified PNA-negative metacyclics of *L. major* MRC-01 (top) and *L. major* MRC-02 (bottom) from stationary phase cultures shown in (**a**). **c–h** In vivo lesion development in BALB/c mice following subcutaneous infection with $10^6$ metacyclics of *L. major* MRC-01 or *L. major* MRC-02 in the presence or absence of 50 mg/kg paromomycin (PM) i.p. daily for 10 days. Treatment was initiated when lesion diameter reached 3–4 mm and mice were killed if lesion size exceeded 9–10 mm in any one direction. Data are presented in aggregated form (**c**, **f**) normalised to the day of treatment initiation (shown as dotted vertical line) and as a timeline for individual mice receiving vehicle alone (**d**, **g**) or PM treatment (**e**, **h**). Data are derived from two independent experiments with 9 mice per group for *L. major* MRC-01 with and without treatment and *L. major* MRC-02 with treatment and $n = 10$ mice for *L. major* MRC-02 without treatment. Data are shown as mean ± SD. Data points within horizontal shaded area represent lesion was palpable but not measurable at <1 mm diameter. Source data are provided as a Source data file.

h and 4a–d). Real-time PCR quantification of parasite kDNA in the lesion indicated that PM treatment reduced parasite load for both strains by >99% (Fig. 4e). Thus, while both strains are capable of causing lesions in BALB/c mice which can be cured using PM, *L. major* MRC-02 demonstrated more rapid and reproducible lesion development.

**Parasite development in the sand fly vector.** To determine whether *L. major* MRC-01 and *L. major* MRC-02 were fully competent for sand fly transmission, we first conducted artificial membrane feeding experiments using two vector species, *Phlebotomus papatasi* and *P. duboscqi*. Experimental infections indicated that both strains developed well in the two sand fly species (Fig. 5a, b), producing high infection rates (100% of sand fly females infected by day 3 post blood meal (PBM), >75% at days 6 and 15 PBM). In *P. duboscqi*, development was more vigorous, with parasite escape from the peritrophic matrix and colonization of the thoracic midgut, cardia and in some cases the stomodeal valve by day 3 PBM. In contrast, in *P. papatasi* the first colonization of the stomodeal valve was not observed until day 6 PBM (Fig. 5c, d). Nevertheless, by day 15 PBM, both sand fly species had supported full development of parasites, with heavy parasite loads in the thoracic midgut and colonization of the stomodeal valve in all the female sand flies infected with both *L. major* strains.

To more precisely quantify parasite load at each morphological stage, exact numbers of procyclic and metacyclic forms in infected sand fly females were counted using a Burker chamber. The differences between *Leishmania* strains were not significant, indicating that both vectors were capable of supporting parasite

development. There was a trend for greater numbers of metacyclic parasites in sand flies infected with *L. major* MRC-02 at day 3 PBM, but this was not apparent by day 15 PBM, with metacyclic numbers ranging from 200 to 258,000 per sand fly (Fig. 6 and Supplementary Data 1). Given recent data suggesting that additional blood meals may serve to enhance the development of metacyclics[45], we conducted a pilot experiment in which we provided sand flies either one additional blood meal on an uninfected BALB/c mouse at day 6 or two additional blood meals at day 6 and day 12 PBM. Under the conditions used, we found no significant differences in metacyclic numbers in *P. duboscqi* infected with either *L. major* strain using these different feeding conditions (Supplementary Data 1). Although both vector species could therefore be suitable for use in a CHIM, the additional robustness of *P. duboscqi*, and a trend towards more permissive parasite development (ref. [52] and this paper) favour use of this species.

To evaluate whether expansion under GMP conditions might affect parasite development, we repeated these experiments using parasites expanded as a research bank (RB) under conditions identical to that for proposed GMP manufacture. Given the data above, we limited these experiments to *P. duboscqi* given a single infectious blood meal by membrane feeding. As before, both *L. major* strains produced 100% infection rates in sand fly females on day 3 PBM, with >80% late-stage infections on day 6 and day 15 PBM. At day 3 PBM, 14% and 43%, respectively, of females infected with *L. major* MRC-01 and *L. major* MRC-02 had parasites located at the stomodeal valve. By day 15 PBM, thoracic midguts were filled with high numbers of parasites and the stomodeal valve was colonized in all the females infected with both strains, though heavier infections developed in sand flies

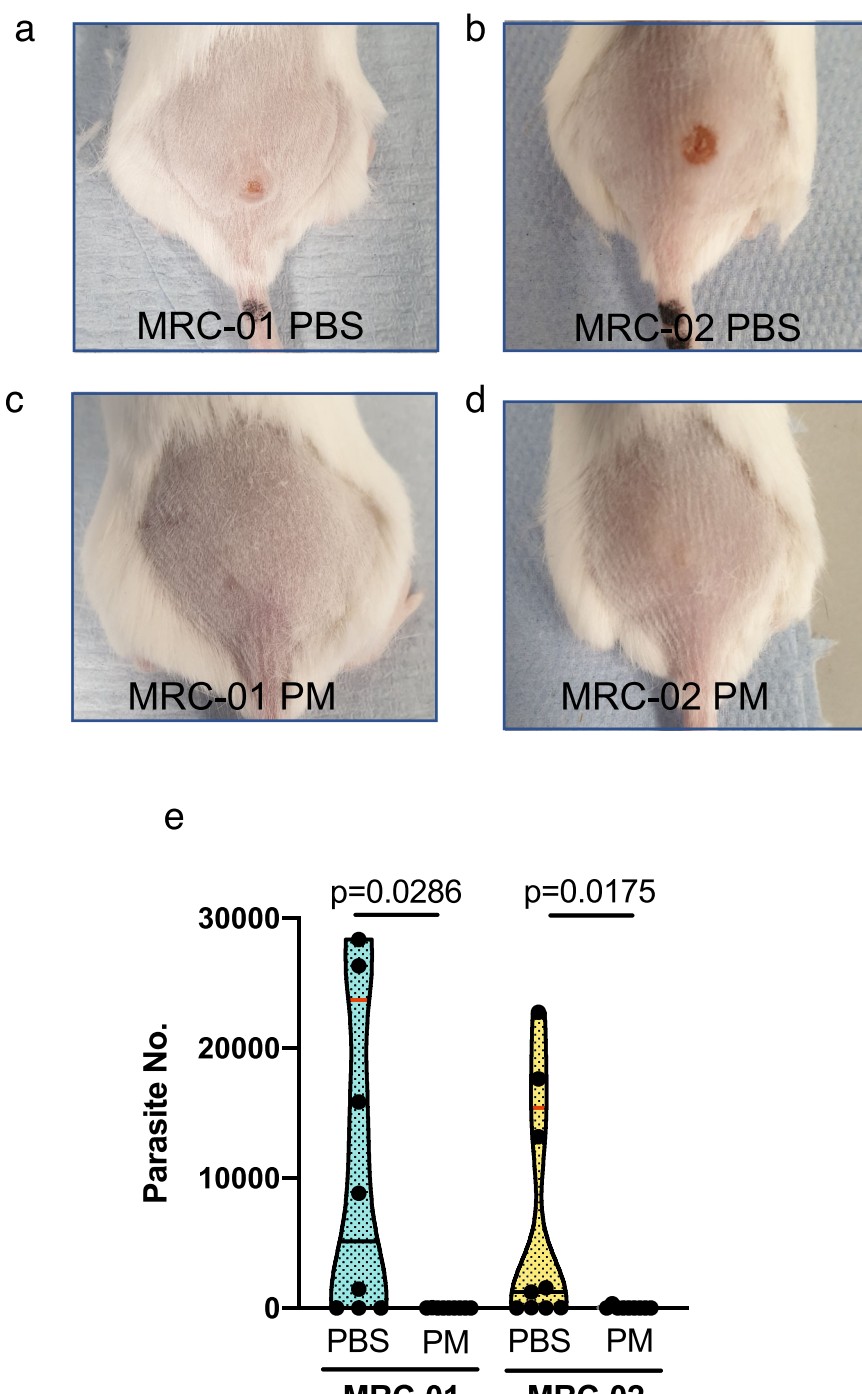

**Fig. 4 Response of BALB/c mice infected with *L. major* MRC-01 and *L. major* MRC-02 to paromomycin treatment.** Representative photographs of BALB/c mice infected with *L. major* MRC-01 (**a**, **c**) and *L. major* MRC-02 (**b**, **d**) in absence of treatment (**a**, **b**) and at the end of 10 days paromomycin treatment (**c**, **d**). **e** Parasite loads in lesions of mice treated with paromomycin (PM) or vehicle (PBS), as determined by qPCR. Data are derived from two independent experiments with 9 mice per group for *L. major* MRC-02 with and without treatment and *L. major* MRC-01 with treatment and $n = 8$ mice for *L. major* MRC-01 without treatment. Data are shown as violin plots truncated at the max/min values, with median (black line) and quartiles (red line) and individual data points indicated. Data comparing with and without treatment were analysed using a two-sided Mann–Whitney test, with *p* values as indicated. Source data are provided as a Source data file.

infected with *L. major* MRC-02 (Supplementary Fig. 5). Thus, the limited expansion required to generate a GMP parasite bank does not negatively impact on parasite development in sand flies.

**Transmission of *L. major* MRC-01 and *L. major* MRC-02 to mice by sand fly bite.** Ten *P. duboscqi* females infected by *L.*

*major* MRC-01 or *L. major* MRC-02 were allowed to feed on anaesthetized BALB/c mice on day 15 post BM. Immediately post feeding, six ear samples per each strain were taken for determination of transmitted parasite number using qPCR. Positivity rates were 5/6 and 6/6 for *L. major* MRC-01 and *L. major* MRC-02, respectively, and numbers of parasites per ear varied from 0 to

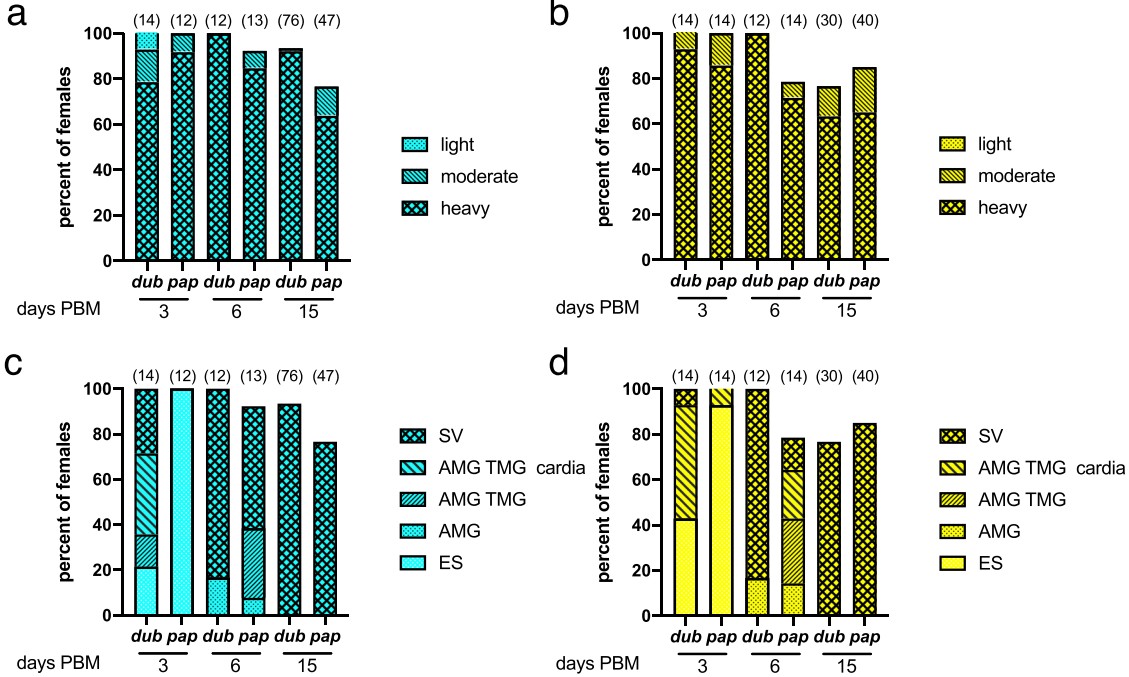

**Fig. 5 Qualitative analysis of *L. major* MRC-01 and *L. major* MRC-02 development in sand flies.** At the days indicated post blood meal (PBM), engorged *P. duboscqi* (*dub*) and *P. papatasi* (*pap*) were dissected and parasite development was assessed. Percentage of flies infected with *L. major* MRC-01 (**a**) and *L. major* MRC-02 (**b**) based on assessment of intensity of infection. Development of *L. major* MRC-01 (**c**) and *L. major* MRC-02 (**d**) as assessed by location within the gut; endoperitrophic space (ES), anterior midgut (AMG), thoracic midgut (TMG) cardia and stomodeal valve (SV). Data are pooled from three independent experiments and are shown as frequency of total number of sand flies dissected. Number of sand flies dissected is shown above each bar. Source data are provided as a Source data file.

7240 and from 92 to 5670, respectively, for the two strains (Fig. 7a and Supplementary Table 1). The average numbers of transmitted parasites did not differ significantly between the two groups.

An independent group of mice was followed to monitor lesion development. Three weeks p.i., only ear swelling was observed in mice bitten by sand flies infected with *L. major* MRC-01, whereas lesions had developed in 3/5 mice exposed to sand flies infected with *L. major* MRC-02. At the end of the experiment on week 6 p.i., all five mice bitten by *L. major* MRC-02 infected flies showed presence of skin lesions while lesions appeared only in a half of mice (3/6) bitten by *L. major* MRC-01 infected flies (Fig. 7b, Supplementary Table 1 and Supplementary Fig. 6). Determination of parasite load by qPCR at 6 weeks p.i. indicated a trend towards higher numbers of parasites in mice infected with *L. major* MRC-02 ($P = 0.08$, Fig. 7c). Of note, two mice exposed to sand flies infected with *L. major* MRC-01 hosted significant numbers of parasites in the ear ($2.82 \times 10^5$ and $6.44 \times 10^5$) despite the absence of lesions (Supplementary Table 1). Hence, *L. major* MRC-02 produces rapid and reproducible lesions in mice after sand fly transmission.

**GMP production of *L. major* MRC-02**. A GMP clinical lot of *L. major* MRC-02 was produced under contract directly from P1 passage stocks using static T-flask cultures. The clinical lot comprises ~600 vials, each vial containing $2 \times 10^7$ mid-log *L. major* MRC-02 in culture media. Vials are stored at $-145 \pm 10$ °C and we expect shelf life to exceed 5 years. An initial 2 years stability study will be performed at the Vibalogics manufacturing site. Release testing of the batch was discussed and agreed with the UK Medicines and Healthcare products Regulatory Agency (MHRA) and comprises identity (PCR), resuscitation (indicating growth), sterility, endotoxin and pH. We estimate conservatively that in a sand fly CHIM, after retention for stability studies, this

clinical lot will be sufficient for challenge of at least 1200 volunteers. Seed stocks are available for further GMP runs as required.

## Discussion

Controlled human challenge is increasingly viewed as being on the critical path for vaccine development, allowing an early demonstration of efficacy and in combination with appropriately designed Phase I trials, rapid selection/de-selection of candidate vaccines[53]. We have therefore sought to develop a new CHIM, based on best practices derived from other models. Three questions influenced our approach to developing a challenge agent, namely which (i) parasite species, (ii) challenge route and (iii) manufacturing standard?

Addressing the first question, *L. major*, the causative agent of Old World CL lends itself to development as a human challenge agent on a number of counts. First, unlike other species causing CL, e.g. *L. tropica*, *L. mexicana* or *L. (Viannia) braziliensis* and *L. (V.) guyanensis*, where systemic or metastatic spread is commonly documented, lesion development following *L. major* infection is usually localised to the site of sand fly transmission and is most commonly self-healing[5]. Numerous barriers to developing a CHIM model of CL using existing *L. major* strains were identified, including limited information on the provenance of parasites held in depositories or in use in research laboratories. Although CHIM studies per se are not under formal regulatory control in the UK, informal advice from the MHRA emphasised the importance of understanding the clinical history of the challenge agent donor, donor status with regard other human infectious agents (e.g. HIV) and the need to ensure absence of contact with bovine sera potentially contaminated with agents known to cause transmissible spongiform encephalopathies. In the case of *Leishmania*, passage history also represents an additional, but often

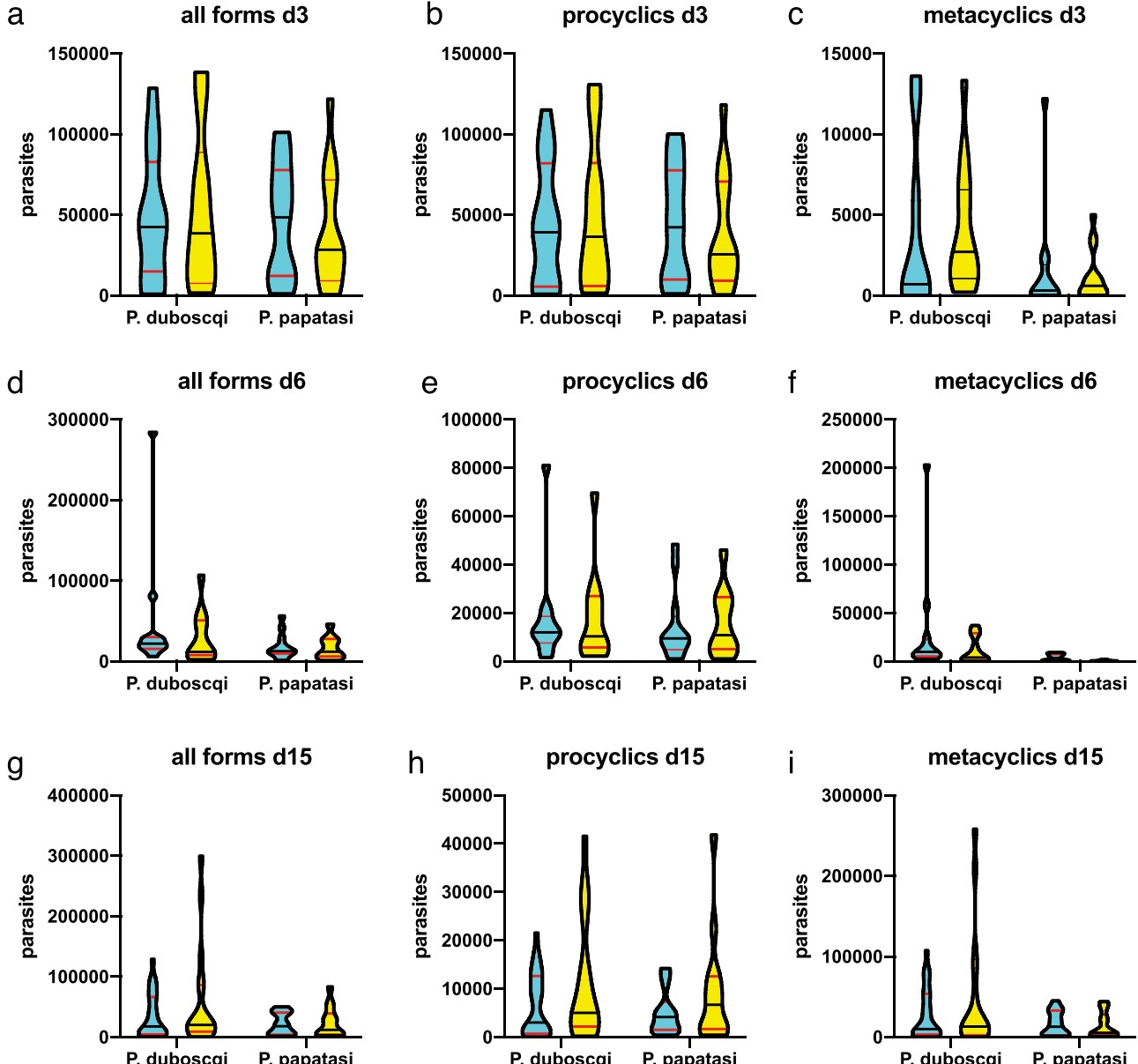

**Fig. 6 Quantitative analysis of *L. major* MRC-01 and *L. major* MRC-02 development in sand flies.** At the days indicated PBM, the number of *L. major* MRC-01 (blue) and *L. major* MRC-02 (yellow) parasites in engorged sand flies was quantified. **a–c** Parasite loads in sand flies at day 3 PBM. **d–f**, Parasite loads in sand flies at day 6 PBM. **g–i** Parasite loads in sand flies at day 15 PBM. Data are shown for all parasites (all forms, **a**, **d**, **e**) and separately for procyclics (**b**, **e**, **h**) and metacyclics (**c**, **f**, **i**). Data are presented as violin plots truncated at the max/min values, with median (black line) and quartiles (red line) indicated and reflect counts obtained from 9 to 25 individual sand flies of each species dissected per time point for each infection. Source data and additional data can be found in Supplementary Data 1.

poorly defined, variable that governs infectivity[54,55]. Hence, two fresh strains were obtained from donors with documented clinical histories and for which we could ensure complete traceability of parasite culture history.

We conducted whole-genome sequencing to establish baseline characteristics and to assess genetic changes occurring after passage in animals. The two parasite strains we examined were genetically distinct, but no features were identified that directly pertain to their potential value as a CHIM agent, given that relatively little is known about the genetic nature of virulence in *Leishmania* parasites. While virulence factors/pathways have been identified in various species, including gp63, lipophosphoglycan, exosome production, proteases and many others[56], how these vary across species or strains and associate with different clinical

presentations is poorly defined[57]. Symbiotic leishmaniaviruses have been associated with enhanced host type I interferon responses and contribute to the metastatic potential in *L. Viannia* species[58]. Although *Leishmaniavirus* has been detected at lower frequency in Old World *L. major* strains[59,60], there is no conclusive data to suggest an involvement in pathogenesis and treatment failure. In Iranian cases, *L. major* infection was not influenced by the presence or absence of LRV2[61]. In any event, both *L. major* MRC-01 and *L. major* MRC-02 were demonstrated to be negative for LRV2.

The classical BALB/c mouse model was used to evaluate in vivo infectivity and drug sensitivity to paromomycin, an often used drug for the treatment of CL[51,62]. Parasite development in two species of sand fly, both natural vectors of *L. major* confirmed full

a
b
c

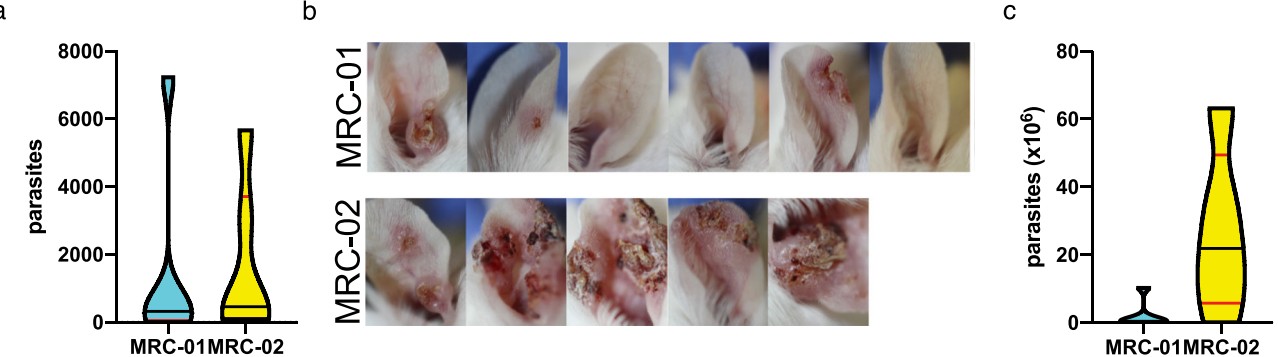

**Fig. 7 Transmission of *L. major* MRC-01 and *L. major* MRC-02 to BALB/c mice by sand fly bite. a** Parasites loads per ear determined by qPCR immediately post bite. Data are shown as violin plot for *n* = 6 ears per parasite strain. Additional data found in Supplementary Table 1. **b** Photographs of ear lesions in individual mice 6 weeks post exposure to sand flies infected with *L. major* MRC-01 and *L. major* MRC-02. For time-course photographs, see Supplementary Fig. 6. **c** Parasite loads determined at 6 week post exposure to infected bites. Data are shown as violin plot for *n* = 5 (*L. major* MRC-02) or 6 (*L. major* MRC-01) ears per parasite strain. Data are derived from a single transmission experiment. Source data are provided as a Source data file.

life cycle completion, including transmission to mice. The latter experiments also pertain directly the second question, route of challenge agent delivery. Vector transmission has been performed for other CHIMs[33,63], and though this approach introduces potential confounding factors such as variability in infectious dose and "take rate" compared to needle injection, this needs to be weighed against the added value of such a model. For leishmaniasis, the body of evidence indicating a synergistic role of sand fly salivary components and parasite secretory products in promoting infectivity[43,44,64,65] coupled with the value of sand fly challenge in identifying vaccine candidates[20,21,46,66] together make a compelling case to proceed with a natural challenge model. The results of a study to optimise a human biting protocol using uninfected sand flies (https://www.clinicaltrials.gov/ct2/show/NCT03999970) and the results of our public involvement activities related to this study will be reported elsewhere.

Finally, we considered whether manufacture should be to GMP or GMP-like, as proposed by others. As it is not possible to generate sand flies that are "GMP", it might have been argued that GMP-like would be sufficient for our purposes. However, given the limited additional costs of producing a clinical lot to GMP and that this leaves open the possibility for direct needle challenge or for any future change in the regulatory framework, we chose to adopt full GMP production of the clinical lot.

We selected *L major* MRC-02 as the challenge agent for GMP production based on an assessment of risk vs reproducibility for participants enrolled in future CHIM studies. The greater consistency in take rate by both needle and sand fly transmission and more rapid lesion development of *L. major* MRC-02 compared to *L. major* MRC-01 has an important bearing on the conduct and management of clinical studies as well as significantly decreasing the number of participants required and the duration of clinical monitoring. For example, in a simple two-arm (placebo vs. vaccine) trial with 90% power to detect a dichotomous outcome (lesion vs. no lesion) at a *p* value of 0.05 and with vaccine efficacy of 60%, a CHIM with 95% take rate would require 24 participants. In comparison, if the take rate was only 60%, the same study would require 74 participants. As additional parasitological and clinical data become available from our early studies using this CHIM model (see below), alternative end points to lesion development may also become valuable as indicators of vaccine efficacy in humans. For example, reducing parasite burden in the skin might impact transmission competence[67] and/or reduced lesion size or rate of evolution may have some quality of life benefit[9]. The GMP clinical parasite bank we have generated is supported by a comprehensive data package (as described here)

and should be sufficient to conduct CHIM studies in >1200 individuals by sand fly transmission. While manufacture was not of the scale associated with development of for example virally-vectored vaccines, it balances yield with the desire to limit in vitro parasite expansion and should serve as a key resource for several years to come.

This study has some limitations. *L major* was chosen as the challenge agent as this represents the species with most limited clinical severity. While the use of a *L major* CHIM would clearly inform the development of prophylactic vaccines against Old World CL, the degree to which this data could be extrapolated to protection against other species, for example *L. donovani* a causative agent of VL is untested currently. Epidemiological data has suggested[68] and experimental evidence supports[69–71] some degree of cross-protection between parasites causing CL and VL, including following vaccination. It is reasonable to suggest therefore that successful protection following vaccination in a *L. major* CHIM would provide highly encouraging albeit not definitive evidence to support the development of vaccines against VL or other forms of CL. Additional limitations of the current study are that it is not possible to accurately predict how infectivity in mice will translate to infectivity in humans and that it is not possible to quantify parasite load in infected sand flies prior to use in a transmission study. To mitigate against these factors, in addition to monitoring parasite load in replicate cohorts of flies alongside those used for human infections studies, we have proposed an adaptive clinical trial design as outlined elsewhere (https://clinicaltrials.gov/ct2/show/NCT04512742). Following ethical approval, we will initially evaluate the frequency of parasitologically-confirmed lesions (take rate), rate of lesion development and parasite load in six volunteers following exposure to five *P. duboscqi* infected with *L. major* MRC-02. As required, the adaptive design allows for variation in the sand fly species, the number of infected sand flies and the biting time. Robust monitoring and early lesion excision will further mitigate against the development of severe disease. The aim of our initial studies will be to optimise the CHIM model and inform the design of subsequent clinical trials employing controlled human infection as a measure of vaccine efficacy. These studies will also provide for detailed mechanistic insights into the early evolution of a primary CL lesion.

## Methods
**Ethics statement**. All human studies were conducted in accord with the Declaration of Helsinki. Ethical approval was obtained from the Helsinki Committees of Hebrew University (0400-18-SOR) and The Chaim Sheba Medical Centre (5658-18-SMC) and the University of York Dept. of Biology Ethics

Committee. Informed consent was obtained from patients with PCR-confirmed leishmaniasis for parasite isolation and subsequent use of these parasites in the development of a human challenge model. Animals were maintained and handled at Charles University and the University of York in accordance with institutional guidelines and national legislation (Czech Republic: Act No. 246/1992 and 359/2012 coll. on Protection of Animals against Cruelty in present statutes at large; UK: Animals (Scientific Procedures) Act 1986). All the experiments were approved by (i) the Committee on the Ethics of Laboratory Experiments of the Charles University in Prague and were performed under permit from the Ministry of Education, Youth and Sports of the Czech Republic (MSMT-28321/2018-6), and (ii) The University of York Animal Welfare and Ethics Review Board and performed under Home Office license (PPL P49487014).

**Mice and parasites.** Adult specific pathogen-free BALB/c mice between 8 and 12 weeks old were used in all experiments reported here and were obtained from either AnLab s.r.o (Prague) or Charles River UK (York). Mice were maintained in individually ventilated cages with food and water ad libitum and a 12 h light/12 h dark photoperiod in rooms maintained at 56% humidity, 20–21 °C. Two new strains, *L. major* MHOM/IL/2019/MRC-01 and *L. major* MHOM/IL/2019/MRC-02 (herein referred to as *L. major* MRC-01 and *L. major* MRC-02), were obtained from patient lesions by culturing in 1 ml Schneider's *Drosophila* medium (Cat. No. 21720024, Gibco) containing 20% foetal bovine serum (Australian origin, Cat. No. 10101145, Lot No. 1951998S, Gibco) at 26 °C. Antibiotics were not included in the medium. The cultures were positive for promastigotes after 9 and 13 days, respectively. The parasites were further expanded and cryopreserved after one and two passages. For freezing, parasites ($2 \times 10^7$ cells/vial) were suspended in Schneider's *Drosophila* medium containing 30% foetal bovine serum and 6.5% DMSO and transferred to a Mr. Frosty box at −80 °C.

**Parasite sequencing and analysis.** DNA from promastigote cultures was extracted using a DNeasy column according to manufacturer's instruction (Qiagen). PCR-free library preparation (Lucigen) and NextSeq 500 sequencing (Illumina) was performed at Genome Quebec. Raw reads were processed as described[72]. Briefly, Illumina paired reads were aligned to the reference *L.major* Friedlin reference genome sequence obtained from TriTrypDB[73] using the Burrows-Wheeler Aligner (version 0.7.17)[74], file formats transformed using samtools (version 1.10), and variant calling was done with VarScan2 (version 2.4.3)[75] to generate VCF files. Per sample candidate SNP were called by VarScan2 with a minimum coverage of 0.4× mean genome coverage, a minimum alternate allele frequency of 20% (read/read), a minimum average base quality of 15 across the reads and a 90% significance threshold. For phylogeny generation, additional sequences obtained from GenBank from whole-genome sequencing projects of *L. major* were also processed and aligned along with *L. major* MRC-01 and *L. major* MRC-02 strains. Polymorphisms and copy number variant were plotted using circos[76] and inspected manually using the Integrative Genomics Viewer[77]. Homozygous SNP comparisons of *L. major* MRC-01 and *L. major* MRC-02 to the *L. major* Friedlin strain were analysed and reported.

*L. major* strains were tested for the presence of LRV2 by RT-PCR using the primers LRV F-HR (5′-tgt aac cca cat aaa cag tgt gc-3′) and LRV R-HR (5′-att tca tcc agc ttg act ggg-3′)[78]. RNA was purified from *L. major* MRC-01, *L. major* MRC-02 or *L. aethiopica* (MHOM/ET/1985/LRC-L494) using the TRI reagent (Sigma-Aldrich) according to the manufacturer's instructions. The latter strain was used as a positive control for LRV2. cDNA was synthesis using the Transcriptor Universal cDNA Master Kit (Sigma-Aldrich) with random hexamer primers. Each PCR reaction (25 µl) contained 5 µl cDNA, 10 µM each primer and 10 µl master mix (PCR-Ready High Specificity, Syntezza Bioscience); and were carried out as follows: Initial denaturation 95 °C for 2 min, 35 cycles at 95 °C for 20 s, annealing at 55 °C for 40 s, extension at 72 °C for 40 s and final extension at 72 °C for 5 min. Amplicons were analysed on 1.5% agarose gels.

**In vivo infectivity by needle challenge.** BALB/c mice were infected s.c. in the shaved rump with 100 µl saline containing $10^6$ metacyclic promastigotes, selected from stationary phase cultures using PNA agglutination[50]. Lesion development was monitored every 2–3 days until patency and daily thereafter. Measurements were performed in two directions using a dial caliper and the mean (8 mm) and maximum single (10 mm) diameter used to evaluate when mice had reached their clinical endpoint. Lesions were collected post mortem, amastigotes allowed to transform into promastigotes (P0) and then frozen as a stock culture (P1) for subsequent sequence analysis (as above). For drug treatments, mice reaching a predetermined cut-off of 4 mm were randomized ($n = 9$–10 per group) to receive either saline or paromomycin (50 mg/kg, i.p. daily for 10 days). Treated mice were killed at day 10 post treatment for evaluation of parasite load by qPCR for kinetoplastid DNA (see below).

**Sand fly colonies and sand fly infections.** The colonies of *P. duboscqi* and *P. papatasi* (originating in Senegal and Turkey, respectively) were maintained in the insectary of the Department of Parasitology, Charles University in Prague, under standard conditions (26 °C on 50% sucrose, humidity in the insectary 60–70% and

14 h light/10 h dark photoperiod)[79]. The sand fly colonies have been screened by RT-PCR and found to be negative for Phleboviruses (including Sandfly Fever Sicilian Virus group, Massilia virus and Toscana Virus) and Flaviviruses (targeting a conserved region of the NS5 gene). Only female sand flies are used in experiments.

Promastigotes from log-phase cultures (day 3–4 in culture) were washed twice in saline and resuspended in heat-inactivated rabbit blood at a concentration of $1 \times 10^6$ promastigotes/ml. Sand fly females (5–9 days old) were infected by feeding through a ethanol-sterilized chick-skin membrane (BIOPHARM) on the promastigote-containing suspension. Engorged sand flies were separated and maintained under the same conditions as the colony. On day 3, 6 and 15 post bloodmeal (PBM) sample sand flies were dissected and digestive tracts examined by light microscopy. Five locations for *Leishmania* infection were distinguished: endoperitrophic space (ES), abdominal midgut (AMG), thoracic midgut (TMG), cardia (CA) and the stomodeal valve (SV). Parasite loads were estimated by two methods: (i) infections were qualitatively assessed in situ as light (<100 parasites per gut), moderate (100–1000 parasites per gut) and heavy (>1000 parasites per gut)[80]; (ii) infections were quantitatively assessed by transferring each gut into 100 µl of 0.01% formaldehyde solution, followed by homogenization and counting using a Burker chamber. *Leishmania* with flagellar length <2 times body length were scored as procyclic forms and those with flagellar length >2 times body length as metacyclic forms[81].

**Sand fly to mouse transmission experiments.** For transmission experiments, BALB/c mice were anaesthetized with ketamin and xylazine (62 mg and 25 mg/kg). Sand flies infected for 15 days (as above) were placed into small plastic tubes covered with the fine mesh (10 females per tube) and the tubes were held on the ear pinnae of anaesthetized mice for 1 h. Engorged sand fly females were immediately dissected for microscopical determination of infection status (as described above). One group of mice was euthanized immediately post transmission and a second group of mice was followed for a period of 6 weeks p.i.

**Determination of parasite load in tissues.** Sand fly-exposed ear pinnae were dissected and stored at −20 °C. Extraction of total DNA was performed using a DNA tissue isolation kit (Roche Diagnostics, Indianapolis, IN) according to the manufacturer's instructions. Lesions from needle challenge were dissected and stored at −80 °C. Extraction of total DNA was performed using DNeasy tissue isolation kit (Qiagen) according to the manufacturer's instruction. Parasite quantification by quantitative PCR (qPCR) was performed in a Bio-Rad iCycler & iQ Real-Time PCR Systems using the SYBR Green detection method (SsoAdvanced™ Universal SYBR® Green Supermix, Bio-Rad, Hercules, CA). Primers targeting 116 bp long kinetoplast minicircle DNA sequence (forward primer (13A): 5′-GTG GGGGAGGGGCGTTCT-3′ and reverse primer (13B): 5′-ATTTTACACCAACCC CCAGTT-3′) were used[82]. One microlitre of DNA was used per individual reaction. PCR amplifications were performed in triplicates using the following conditions: 3 min at 98 °C followed by 40 repetitive cycles: 10 s at 98 °C and 25 s at 61 °C. PCR water was used as a negative control. A series of 10-fold dilutions of *L. major* promastigote DNA, ranging from $5 \times 10^3$ to $5 \times 10^{-2}$ parasites per PCR reaction was used to prepare a standard curve. Quantitative results were expressed by interpolation with a standard curve. To monitor non-specific products or primer dimers, a melting analysis was performed from 70 to 95 °C at the end of each run, with a slope of 0.5 °C/c, and 5 s at each temperature.

**Statistical analysis.** Data are plotted using violin plots and mean, median, 95% CI and ranges are shown as appropriate. Where shown, error bars represent standard deviation of the mean. All statistical analysis was performed with the statistical software package SPSS version 23 or with GraphPad Prism 8 for macOS (v8.4.2). Normality was evaluated using the D'Agostino–Pearson test and differences in parasite numbers in mice and sand fly tissues were tested by non-parametric (Mann–Whitney U, Mood's median test) or parametric tests (student's *t* test or ANOVA) depending on data distribution. Time to event analysis was conducted using the log-rank (Mantel-Cox) test.

**Reporting summary.** Further information on research design is available in the Nature Research Reporting Summary linked to this article.

## Data availability

Sequence data for *L. major* MRC-01 and *L. major* MRC-02 are available from GenBank (BioProject PRJNA633113: accession numbers SAMN14933143, SAMN14933144 and SAMN14933145. Parasite genomics data used in compiling Fig. 2 are available from TriTrypDB (https://tritrypdb.org/tritrypdb/app). Parasites produced under GMP will be available for clinical assessment of candidate *Leishmania* vaccines under an appropriate MTA. Source data and additional data supporting Fig. 6 are provided in Supplementary Data 1. Source data are provided with this paper.

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

## Acknowledgements

The authors thank Shaden Kamhawi (NIAID, NIH), Hira Nagasi (FDA, NIAID), Steve Reed (ONC Bio) and Jesus Valenzuela (NIAID, NIH) for helpful discussions. This work was funded by a Developmental Pathways Funding Scheme award (MR/R014973 to P.M.K., C.L., A.L., P.V. and C.L.J.). This award is jointly funded by the UK Medical Research Council (MRC) and the UK Department for International Development (DFID) under the MRC/DFID Concordat agreement and is also part of the EDCTP2 programme supported by the European Union. P.V. and J.S. were partially supported by European Regional Development Funds (project CePaViP 16_019/0000759).

## Author contributions

H.A. performed in vitro and in vivo needle challenge experiments, analysed data and generated figures. J.V., B.V. and T.B. conducted parasite development studies in sand fly and sand fly transmission experiments. P.L. and G.M. conducted parasite genome analysis. E.S. was the clinician responsible for patient recruitment and care and contributed to the clinical protocol. E.G. was the project manager and data coordinator. K.v.B. conducted in vitro experiments. K.L. and M.P. developed methodology and produced the research and GMP parasite banks. C.J.L. was the sponsor's clinical representative, wrote the clinical protocol and obtained the funding. C.L.J. conducted experiments. P.V. coordinated the sand fly experiments and P.M.K. analysed data and produced the first draft of the manuscript. V.P. and E.G. contributed to project design. A.M.L., C.L., C.L.J., P.V. and P.M.K. conceived the project and obtaining funding. All authors contributed to reviewing the manuscript.

## Competing interests

The authors declare no competing interests. P.M.K. and C.J.L. are co-authors of a patent protecting the gene insert used in candidate vaccine ChAd63-KH (Europe 10719953.1; India 315101).
