## [Peer Review File · Nature Communications]

Reviewer comments first round -

Reviewer #1 (Remarks to the Author):

This well written manuscript provides a methodic, comprehensive genotype and phenotype analyses of two fresh human *Leishmania* major isolates with well documented culture histories, and includes the GMP manufacture of one of the isolates. While the studies provide little in the way of a scientific advance, they represent an essential step in the development of a controlled human infection model that itself will transform the ability to evaluate candidate vaccines against *Leishmania* major and possibly other *Leishmania* species. The genotype analysis confirmed the close identity of the strains to the reference Israeli strain. The absence of an RNA virus described in other *L. major* strains was demonstrated. The ability of the strains to grow and produce lesions in BALB/c mice, their drug sensitivity to paromomycin, their ability to colonize sand flies, differentiate into infectious metacyclic promastigotes, and finally their ability to be transmitted by sand fly bite to produce cutaneous lesions in mice were in each case carefully investigated and the findings are clear. The conclusion that the MRC-02 strain is suitable to be used in future controlled human challenge trials is well supported.

A few points to address:

Please clarify if the identified SNPs comparing the different strains represent homozygous or heterozygous SNP differences, or both.

For WGS of parasites following in vivo passage, were tissue amastigotes used or were the parasites cultured as promastigotes prior to sequencing?

Why were 10 flies per mouse chosen for the transmissions? Is it based on a particular calculus taking into account the average intensity of infection or average number of metacyclics? Was there variability in transmission success in different experiments? How will the number of infected flies be determined in the human exposures? A take rate for MRC-01 of 100% could presumably be achieved if more flies were used. So if there is an important benefit of using a strain that produces less aggressive lesions, then the efficiency of transmission might not need to be the most important variable.

Reviewer #2 (Remarks to the Author):

General comment

The informative and necessary data collected from putative challenge agents for controlled human leishmaniasis infection can be considered routine in *Leishmania* research laboratories. Methodology cannot be considered innovative, and results were almost expected. Therefore it is wondered if the manuscript submission to a pure scientific journal meets the objective of this important preliminary work on efficacy standardization of vaccines.

Specific comments

L53: please include again the concept that the type of disease (or the outcome of infection) largely depends on parasite species, which sounds equally (if not more) important than parasite species co-evolution with sand flies taxa. Furthermore, a mention to the high prevalence of asymptomatic infections in endemic settings of VL may be important to further highlight problems encountered in large-scale efficacy studies of vaccines.

L69-70: please consider that the marketed canine leishmania vaccines were developed by exposure of dogs to the natural transmission in settings endemic for the homologous parasite. These vaccines do not have an impact on the prevalence of acquired infection (as determined by

conventional and quantitative PCR), but on the development of clinical disease. This information should be mentioned in Introduction and discussed later in Discussion, regarding the expected level of protection conferred by human vaccines.

[L69; L83: parentheses are missing]

L121: Would it be possible to label these strains also according to the WHO standard code for Leishmania isolates, along with the short donor/lab code? This is quite informative as it reports on the nature of the host, country and year of isolation, with no need to read back this manuscript in the future

L187: why "highly susceptible"? Balb/c mice are known to be susceptible to L. major infection, did you use a sub-strain of mice more susceptible compared with the average?

L202: are we sure that a rapid lesion development is a preferred feature for an agent (e.g. L. major MRC02) to be used in CHIM? What about the risk of developing a severe lesion if not properly and timely controlled? On the other hand, see L266 regarding sand fly-transmitted infection without apparent lesion, obtained with MRC01 which might represent a safer parasite. The discussion about this issue (from L347) is unconvincing, as a modern phase II/III investigation for efficacy of human vaccines should be able to explore parameters different from the old "clinical lesion vs no lesion" (a parasite DNA burden, for example). Or at least should tend to it in the future.

L405: please check if "smears" is the correct word, as this preparation dries very quickly being optimal for parasite staining but not for its survival in culture.

L460: does the use of chicken skin requires particular safety issues? Are there viruses or bacteria in this membrane that can be ingested by sand flies before assuming the Leishmania-infected blood?

NCOMMS-20-30698
Response to reviewers

Reviewer #1:

This well written manuscript provides a methodic, comprehensive genotype and phenotype analyses of two fresh human Leishmania major isolates with well documented culture histories, and includes the GMP manufacture of one of the isolates. While the studies provide little in the way of a scientific advance, they represent an essential step in the development of a controlled human infection model that itself will transform the ability to evaluate candidate vaccines against Leishmania major and possibly other Leishmania species. The genotype analysis confirmed the close identity of the strains to the reference Israeli strain. The absence of an RNA virus described in other L. major strains was demonstrated. The ability of the strains to grow and produce lesions in BALB/c mice, their drug sensitivity to paromomycin, their ability to colonize sand flies, differentiate into infectious metacyclic promastigotes, and finally their ability to be transmitted by sand fly bite to produce cutaneous lesions in mice were in each case carefully investigated and the findings are clear. The conclusion that the MRC-02 strain is suitable to be used in future controlled human challenge trials is well supported.

We thank the reviewer for recognising the importance and transformative nature of this work and for the compliments on the quality of the manuscript.

1. Please clarify if the identified SNPs comparing the different strains represent homozygous or heterozygous SNP differences, or both.

With respect to Figure 2B, the comparison of *L. major* MRC-01 and *L. major* MRC-02 with Friedlin (Grey bars) are all homozygous polymorphisms. The comparison of *L. major* MRC-02 post infection with Friedlin (Red bars) are heterozygous polymorphisms. This is now stated in the revised text (**line 159, 167-169 and 441-442; Figure 2 legend**). We have also amended the table in Figure S1 (**revised Figure S1**) to indicate that the polymorphisms are heterozygous.

2. For WGS of parasites following in vivo passage, were tissue amastigotes used or were the parasites cultured as promastigotes prior to sequencing?

The WGS was performed on promastigotes. Lesions were collected post mortem, amastigotes allowed to transform into promastigotes (P0) and then frozen as a stock culture (P1) for subsequent WGS analysis. We have added this information to the Methods (**lines 463-465**)

3. Why were 10 flies per mouse chosen for the transmissions? Is it based on a particular calculus taking into account the average intensity of infection or average number of metacyclics? Was there variability in transmission success in different experiments? How will the number of infected flies be determined in the human exposures? A take rate for MRC-01 of 100% could presumably be achieved if more flies were used. So if there is an important benefit of using a strain that produces less aggressive lesions, then the efficiency of transmission might not need to be the most important variable.

We used 10 flies in mouse transmission experiments, as neither the parasite infection rate nor the feeding rate of sand flies on mouse ears were 100% and the number of infected engorged females ranged mostly between 2-4 females per mouse (Table S2). The mouse transmission studies were conducted in a single run. We believe this was appropriate to fulfil the aim of these experiments, namely to confirm that the parasite strains underwent full life cycle development and were sand fly transmissible and giving consideration to the ethical use of animals. The number of repeats has been added to the legend (**revised Figure 7 legend**).

For human exposures, reducing risk and inconvenience to the volunteers, maximising the consistency of the end point and having a manageable clinical protocol are of paramount importance. In the human studies, sand flies fed on infected blood will be monitored for infection by microscopy or PCR. Our clinical protocol (currently under National Research Ethics committee review) includes an adaptive design, informed by a pilot study using uninfected sand flies (Parkash et al, ms in preparation; FLYBITE, NCT03999970) that indicates that five sand flies per biting chamber is sufficient to ensure 100% of participants receive at least one bite. We will initially expose 6 volunteers to infected *P. duboscqi*. Should take rates be below 4/6, we will use *P. papatasi* in a further 6 volunteers. Further adaptation covered by the protocol includes increasing or decreasing the number of sand flies used or the time for biting. The main aim here is to achieve a suitably high take rate, whilst minimising risk (e.g. of multiple lesion development) through exposure to unnecessarily high numbers of infected sand flies.

In addition to considering take rate in mice, we also considered *L. major* MRC-02 most suitable for use based on the reproducibility of lesion development by both needle and sand fly bite. Asynchronous lesion development, as seen with *L. major* MRC-01, would be much more difficult to manage clinically and in terms of defining study end points. We have expanded the discussion to include some of these points (**lines 351-355, 359-364 and 380-394**).

Although in the sand fly transmission experiment we did not follow lesion development beyond 6 weeks (due to the imposition of covid-19 restrictions), we feel that it is likely (based on the needle infection data in Figure 3) that *L. major* MRC-01 lesions would have progressed further in time. We have therefore removed the description of *L. major* MRC-02 as being “more aggressive”.

Reviewer #2

General comment: The informative and necessary data collected from putative challenge agents for controlled human leishmaniasis infection can be considered routine in Leishmania research laboratories. Methodology cannot be considered innovative, and results were almost expected. Therefore it is wondered if the manuscript submission to a pure scientific journal meets the objective of this important preliminary work on efficacy standardization of vaccines.

We agree with the reviewer that many of the experimental approaches are standard, though we suspect few groups in the world would be able to complete the spectrum of studies (including sand fly transmission) that are reported here. Nevertheless, we would stress the importance and essential nature of this work for progressing vaccine development.

L53: please include again the concept that the type of disease (or the outcome of infection) largely depends on parasite species, which sounds equally (if not more) important than parasite species co-evolution with sand flies taxa. Furthermore, a mention to the high prevalence of asymptomatic infections in endemic settings of VL may be important to further highlight problems encountered in large-scale efficacy studies of vaccines.

We agree with the reviewer on these points and have included additional text and references in the Introduction (**lines 53-54 and 77-78**)

L69-70: please consider that the marketed canine leishmania vaccines were developed by exposure of dogs to the natural transmission in settings endemic for the homologous parasite. These vaccines do not have an impact on the prevalence of acquired infection (as determined by conventional and quantitative PCR), but on the development of clinical disease. This information should be mentioned in Introduction and discussed later in Discussion, regarding the expected level of protection conferred by human vaccines.

The reviewer makes some good points and we have referenced canine leishmaniasis vaccine studies in the Introduction (**lines 70-72**) and also added comments on outcome measures in human vaccine studies to the Discussion (**lines 359-364**).

[L69; L83: parentheses are missing]

Thank you for pointing out the editorial errors. These have been corrected.

L121: Would it be possible to label these strains also according to the WHO standard code for Leishmania isolates, along with the short donor/lab code? This is quite informative as it reports on the nature of the host, country and year of isolation, with no need to read back this manuscript in the future

This is a very valuable suggestion and we have now included the WHO reference name in the Methods (**lines 421-422**) and on first use in the Results section (**lines 142-144**), whilst using the shorthand terminology of *L. major* MRC-01 and *L. major* MRC-02 elsewhere throughout the manuscript. We have also consulted with Patrick Bastien at the WHO Leishmania Reference Laboratory in Montpellier, and on his advice used the term “strain” rather than “isolate” throughout the manuscript.

L187: why “highly susceptible”? Balb/c mice are known to be susceptible to L. major infection, did you use a sub-strain of mice more susceptible compared with the average?

The experiments were performed with standard commercially available BALB/c mice. We have removed the word “highly” to avoid any confusion.

L202: are we sure that a rapid lesion development is a preferred feature for an agent (e.g. L. major MRC02) to be used in CHIM? What about the risk of developing a severe lesion if not properly and timely controlled? On the other hand, see L266 regarding sand fly-transmitted infection without apparent lesion, obtained with MRC01 which might represent a safer parasite. The discussion about this issue (from L347) is unconvincing, as a modern phase II/III investigation for efficacy of human vaccines should be able to explore parameters different from the old “clinical lesion vs no lesion” (a parasite DNA burden, for example). Or at least should tend to it in the future.

The reviewer makes valuable points that we hope we have clarified in the revised discussion (**lines 351-355, 359-364 and 380-394**). It is of course not possible to predict from animal studies what the rate or course of lesion development will be in human volunteers. As discussed in response to R1 above, safety, compliance, clinical management of the study and ability to terminate the infection are all factors that have been considered. In terms of the chances of developing a severe lesion, in our trial protocol there is continual monitoring of lesion development in clinic and through home monitoring, and we will be terminating lesion development at \geq 3mm by excision. The decision to use *L. major* MRC-02 was made by the investigators in discussion with the Scientific Advisory Group (Valenzuela, Kamhawi, Reed, Naghasi) based primarily on the reproducibility in lesion development by needle or sand fly bite. An adaptive trial design will be employed to maintain flexibility as a CHIM study progresses. Lesion development is however a suitable and well defined clinical end point with which to initiate these studies. It is not currently known to what extent parasite numbers in the lesion will vary following sand fly bite but such information will be collected in the first study as a secondary endpoint and may inform more refined endpoints. We can assure the reviewer that volunteers in any CHIM study will be monitored using state-of-the-art methods to gain the maximum possible parasitological and immunological data from their participation.

L405: please check if “smears” is the correct word, as this preparation dries very quickly being optimal for parasite staining but not for its survival in culture.

We have omitted this term for simplicity.

L460: does the use of chicken skin requires particular safety issues? Are there viruses or bacteria in this membrane that can be ingested by sand flies before assuming the Leishmania-infected blood?

The chick skins are sterilized with ethanol and stored at -20C before use. We have added this information to the text (**line 481**).

Reviewer comments second round -

Reviewer #1 (Remarks to the Author):

All of the substantive comments have been adequately addressed in the revised manuscript and in the author responses.